# Deep learning-based forest fire detection using an improved SSD algorithm with CBAM

**Diansheng Zhang**[1], **Yueyuan Zhang**[1*], **Leilei Dong**[2], **Shifeng Ruan**[3], **Zhiwei Liu**[1]

**1** School of Information and Software Engineering, East China Jiaotong University, Nanchang, China, **2** School of Communications and Information Engineering, Xi'an University of Posts and Telecommunications, Xian, China, **3** School of Flight, Anyang Institute of Technology, Anyang, China

\* zyyaney1981@hotmail.com

## Abstract

Fires are characterized by their sudden onset, rapid spread, and destructive nature, often causing irreversible damage to ecosystems. To address the challenges in forest fire detection, including the varying scales and complex features of flame and smoke, as well as false positives and missed detections caused by environmental interference, we propose a novel object detection model named CBAM-SSD. Firstly, data augmentation techniques involving geometric and color transformations are employed to enrich the dataset, effectively mitigating issues of insufficient and incomplete data collected in real-world scenarios. This significantly enhances the SSD model's ability to detect flames, which exhibit highly variable morphological characteristics. Furthermore, the CBAM module is integrated into the SSD backbone network to reconstruct its feature extraction structure. This module adaptively weights flame color and smoke texture along the channel dimension and highlights critical fire regions in the spatial dimension, substantially improving the model's perception of key fire features. Experimental results demonstrate that the CBAM-SSD model is lightweight and suitable for real-time detection, achieving a mAP@0.5 of 97.55% for flames and smoke, a 1.53% improvement over the baseline SSD. Specifically, the AP50 for flame detection reaches 96.61%, a 3.01% increase compared to the baseline, with a recall of 96.40%; while the AP50 for smoke detection reaches 98.49%, with a recall of 98.80%. These results indicate that the improved model delivers higher detection accuracy and lower false and missed detection rates, offering an efficient, convenient, and accurate solution for forest fire detection.

## 1. Introduction

Forest fires are catastrophic events with global impact, capable of causing severe damage to human life and property in a very short time. These fires often occur in remote and topographically complex mountainous regions, where the combination of challenging terrain and isolated locations makes it difficult for personnel to conduct

**Data availability statement:** All relevant data are within the paper and its Supporting Information files.

**Funding:** This study was supported by the Graduate Innovative Special Fund Projects of Jiangxi Province in the form of a grant awarded to D.Z. (YC2025-S133) and the Anyang City Science and Technology Research Project in the form of a grant awarded to S.R. (2025C01GX036). The funders had no role in study design, data collection and analysis, decision to publish, or preparation of the manuscript.

**Competing interests:** The authors have declared that no competing interests exist.

comprehensive inspections during routine patrols. With the rapid advancement of artificial intelligence technologies, object detection has emerged as a promising approach for addressing this issue. Automated detection systems based on object detection algorithms have increasingly become an effective means of forest fire prevention [1]. Intelligent detection devices, characterized by large data throughput, rapid information transmission, and high recognition accuracy, can effectively curb the rapid spread of flames. High-precision detection and real-time alerts for forest fires are of great significance for the protection of ecological resources.

Object detection is a key subfield of computer vision. With the increasing adoption of computer vision technologies, significant progress has been made in various domains such as surveillance systems, medical diagnostics, and image retrieval [2]. Traditional object detection algorithms, which relied heavily on manually extracted features, suffered from limited adaptability and poor performance. The rapid advancement of deep learning—characterized by its speed, scalability, and end-to-end learning capabilities—has led to the emergence of a new generation of object detection algorithms. The rise of Convolutional Neural Network (CNN) [3] has undoubtedly elevated object detection to a new level. CNN-based detection algorithms have become prevalent due to their ability to leverage large datasets, powerful computational resources, and complex logical structures, supporting broad application scenarios. These algorithms can be broadly categorized into anchor-based and anchor-free methods. Among anchor-based models, research primarily focuses on one-stage and two-stage architectures. YOLO (You Only Look Once) [4] is a representative one-stage object detection algorithm that processes the entire image in a single forward pass. It adopts a unified regression-based loss function and utilizes a single-network architecture, enabling high-speed detection. However, this global approach can sometimes compromise localization precision, leading to a higher risk of false positives. The Single Shot MultiBox Detector (SSD) [5] discretizes the output space with multiple aspect ratios and combines features at different resolutions to predict categories efficiently in a single step, making it more easily integrable. Both YOLO and SSD are typical one-stage detection algorithms. In contrast to one-stage detectors, two-stage methods such as Faster Regions with Convolutional Neural Networks Features (Faster R-CNN) [6] and Spatial Pyramid Pooling Network (SPPNet) [7] first generate region proposals through selective search or a Region Proposal Network (RPN). These proposals are then refined through subsequent classification and bounding box regression. Although these methods offer high detection accuracy, they are computationally intensive and time-consuming, making them less suitable for real-time applications. One-stage detectors, by applying the full convolutional network directly to the detection task, provide a simpler architecture with superior real-time performance.

Traditional fire detection methods primarily rely on features such as flame color, temporal correlation between consecutive frames, and the density characteristics of early-stage smoke. These features are often combined with CNN architectures for detecting fire-related targets. Alexey Bochkovskiy and colleagues [8], aiming to maintain model lightweightness, adopted the YOLOv4-Tiny algorithm. To address

the issue of high missed detection rates, they introduced multi-scale prediction and utilized a binary K-Means clustering algorithm [9] to generate anchor boxes. The model was trained using pretrained weights, enabling high detection accuracy while maintaining fast inference speed and reducing the missed detection rate. Bose SR et al. [10] proposed an improved SSD-based model that replaces the standard backbone with the Extreme Inception (Xception) network, which is built upon depthwise separable convolutions. To compensate for performance degradation caused by lightweight design, they incorporated a Local Attention Module (LAM) to enhance detection performance. Huo B and colleagues [11] replaced the traditional Visual Geometry Group (VGG-16) backbone with a Residual Network (ResNet)- based architecture [12], and introduced a Feature Pyramid Network (FPN) for multi-scale feature fusion, effectively addressing the limitations of SSD in detecting small targets. Qingbang Shi and others [13] optimized the YOLOv2-Tiny model to improve solution search efficiency, and proposed an integrated system combining Unmanned Aerial Vehicles (UAVs) with gimbal tracking to overcome geographic constraints in fire detection. Chen H and others [14] applied transfer learning to improve the VGG-16 backbone network and used data augmentation techniques to expand the dataset. The enhanced model achieved higher classification accuracy and enabled faster and more accurate fire information acquisition.

Above fire detection algorithms, due to increasingly deep network structures and architectural complexity, often result in highly intricate parameter distributions. This poses challenges for real-time deployment and increases the risk of missed detections, particularly for small targets. To address these issues, this study proposes CBAM-SSD, an object detection algorithm based on the SSD framework, aiming to improve detection performance and enable comprehensive monitoring of forest areas. The improved algorithm achieves superior detection accuracy and efficiency compared to most mainstream detection models.

The structure of this paper is as follows: The Section 2 introduces the fundamental principles of the VGG-16-based SSD, the applied data augmentation strategies, the structure of the improved Convolutional Block Attention Module (CBAM) module, and the rationale behind its integration. It also analyzes the specific characteristics of fire detection, identifies the limitations of SSD, and discusses the advantages of using CBAM. The Section 3 describes the custom dataset, experimental settings, training parameters, evaluation metrics, convergence analysis, and presents comparative results and analysis. The Section 4 concludes with a summary of key findings and outlines directions for future research.

## 2. Algorithm principle

### 2.1 VGG-16-SSD target detection algorithm

SSD-Net (Single Shot MultiBox Detector Net) utilizes VGG-16 [15] as its backbone network and extends the original VGG structure by adding convolutional layers along with two fully connected layers. The overall architecture is illustrated in Fig 1. The algorithm performs hierarchical feature extraction and applies bounding box encoding and decoding. At the input stage, images are resized to a fixed resolution. Then, six feature maps at different scales are extracted through convolutional operations for prediction. An anchor-box-like mechanism is employed to preset prior boxes with various aspect ratios on multi-scale feature maps. Based on the effective receptive field, the label positions are determined to complete the classification and localization tasks. Each grid cell outputs offset values and the corresponding confidence scores for each category. Shallow and deep feature maps represent different levels of semantic information and are used to detect objects at different scales. The overall loss function combines classification loss and localization loss through weighted summation. To suppress redundant bounding boxes, Non-Maximum Suppression (NMS) [16] is applied during the post-processing stage. During training, an imbalance between positive and negative samples may arise due to anchor box matching. To address this, SSD employs Hard Negative Mining (HNM) [17], which sorts negative samples based on confidence error and selects low-confidence samples as negatives to ensure a balanced training set.

After pretraining, it was observed that the SSD algorithm based on VGG-16 becomes relatively bulky and less portable due to repeated convolutional operations. The model also exhibits slower convergence and tends to lose information related to small objects. Moreover, the shallow convolutional layers lack sufficient depth, leading to inadequate feature

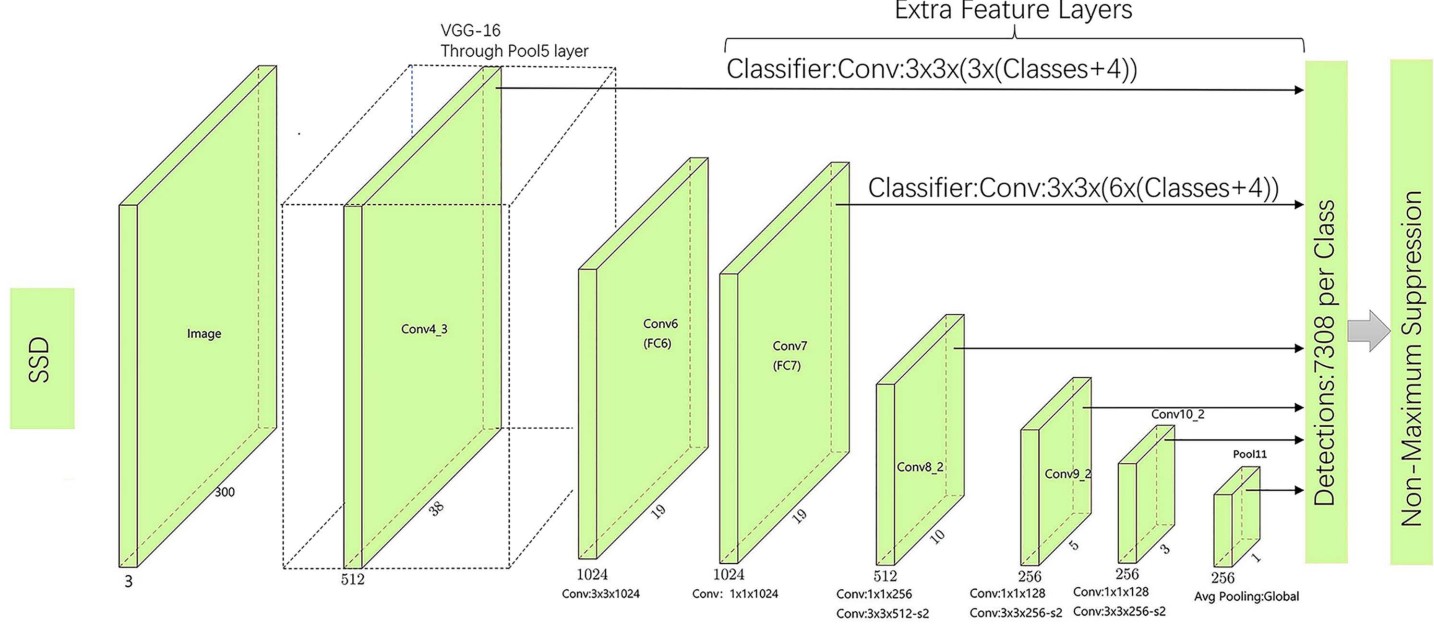

**Fig 1. Architecture diagram of the VGG-16-SSD algorithm.**

extraction for small targets and resulting in weak representational capacity. Consequently, the SSD model is prone to missed detections. Despite its limitations, SSD remains a regression-based framework with high localization accuracy and a favorable balance between recall and model efficiency. Building upon these strengths, this study proposes an enhanced SSD-based algorithm designed to improve small-object detection performance while reducing computational overhead for real-world deployment.

## 2.2 Data augmentation

High-quality data plays a crucial role in model training, directly influencing both the effectiveness of the training process and the model's generalization capability. Due to the limited availability of public datasets, this study utilizes a multi-scenario fire dataset constructed by the laboratory. The data were collected from real fire incidents, film footage, and fire simulations conducted in open environments. The dataset features a wide variety of perspectives, scenes, and flame patterns. However, the data collection process was affected by environmental conditions. The image samples include low-quality instances characterized by blurred edges, image distortion, and blocky artifacts. Additionally, many images contain excessive redundancy, noise, missing values, or lack of prominent regions of interest. Machine learning is inherently complex, and supervised learning relies heavily on feeding the network with accurately labeled data, which provides clear learning targets and enhances the network's ability to extract task-relevant features. Nevertheless, limitations in data availability, the high cost of manual annotation, and insufficient dataset scale hinder adequate model training, thereby restricting its ability to reach its full performance potential.

To avoid the network becoming overly dependent on certain attributes, which can reduce its generalization ability, this study expands the dataset by augmenting images with distinct features. This helps improve the feedback of meaningful information, enrich the semantic content of the data, and enhance both the network's fitting capacity and the model's robustness [18]. In this study, smoke and flame data are combined for training. As smoke features are generally more distinct and easier to extract than flame features, the performance metrics for smoke detection tend to reach the required

thresholds more reliably. Considering that smoke often precedes the visible onset of fire, accurate smoke detection plays an equally critical role in fire prevention. To improve detection performance, offline data augmentation is applied to both smoke and flame samples [18], including geometric and image-based transformations, as shown in Fig 2. Geometric transformations modify the target's spatial position within the original image by operations such as flipping, mirroring, random cropping, and scaling. Image-based transformations involve pixel-level grayscale adjustments in the color, spatial, or frequency domains, and are implemented through RGB conversion, or by adding Gaussian white noise or salt-and-pepper noise. Although the fundamental features represented by the images remain unchanged, for an under-trained network, the augmented images convey semantic content that differs from the original images. This allows the network to relearn features from these enhanced samples, facilitating improved representation learning.

### 2.3 Lightweight Attention Mechanism

The attention mechanism is a specialized component integrated into machine learning models to enable selective focus within the network. By incorporating attention, the model becomes capable of emphasizing more relevant features. Therefore, the improved SSD network introduces a lightweight CBAM [19] into its architecture, as illustrated in Fig 3. The CBAM module focuses on two separate dimensions: channel and spatial, and performs adaptive feature refinement in a

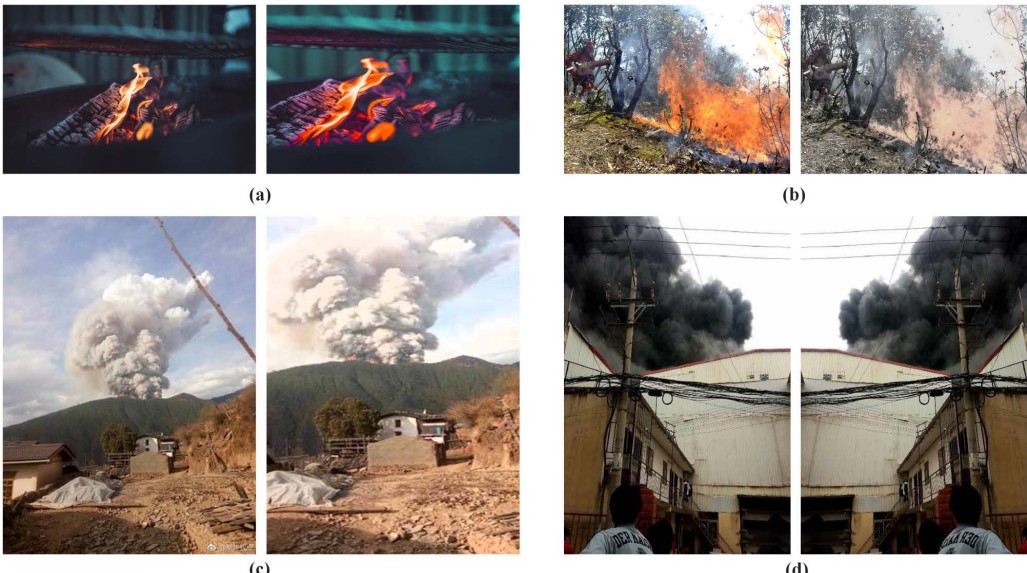

**Fig 2. Data-enhanced results.** (a) Color change. (b) Grayscale transformation. (c) Size change. (d) Mirror transformation.

### Convolutional Block Attention Moudle

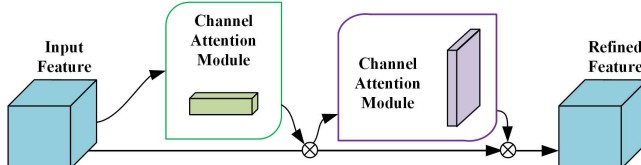

**Fig 3. CBAM structure diagram.**

sequential manner. As a lightweight and plug-and-play module, CBAM can be seamlessly embedded into standard convolutional neural network architectures, offering improved interpretability. The network applies Global Average Pooling (GAP) and Global Max Pooling (GMP) [20] sequentially within different submodules to adaptively refine the input feature maps, which enhances feature representation and improves learning efficiency.

The structure and formulation of the channel attention module are shown in Fig 4. Given an input feature map $F \in R^{H \times W \times C}$ , channel attention is computed by applying both GAP and GMP operations in parallel, each producing a $1 \times 1 \times C$ feature map. These operations serve to capture different spatial semantic descriptors. The two resulting feature maps are then passed through a shared multilayer perceptron (MLP) [21]. The output vectors are fused via element-wise addition after a Sigmoid activation [22], resulting in the refined channel attention feature map $F'$ as defined in Equation (1).

$$\begin{aligned} F' = M_c(F) &= \sigma(MLP(AvgPool(F)) + MLP(MaxPool(F))) \\ &= \sigma(W_1(W_0(F^c_{avg})) + W_1(W_0(F^c_{max}))) \end{aligned}$$
$$(1)$$

The structure and formulation of the spatial attention module are presented in Fig 5 and Equation (2), respectively. This module operates on the refined feature map $F'$ generated by the channel attention module and performs spatial compression to reduce dimensionality. GAP and GMP are applied along the channel axis to extract semantic information from different perspectives, and the resulting feature maps are concatenated. A convolutional layer with a $7 \times 7$ kernel is then applied to expand the receptive field, thereby enriching the spatial positional information.

$$\begin{aligned} M_s(F) &= \sigma(f^{7 \times 7}([AvgPool(F); MaxPool(F)])) \\ &= \sigma(f^{7 \times 7}([F^s_{avg}; F^s_{max}])) \end{aligned}$$
$$(2)$$

The Squeeze-and-Excitation Networks (SENet) module [23] is lightweight, flexible, and contains relatively few parameters, making it suitable for integration into standard neural network architectures. The SE block adaptively learns relative feature weights based on the model's loss during training and applies corresponding reweighting, enabling the network

**Channel Attention Module**

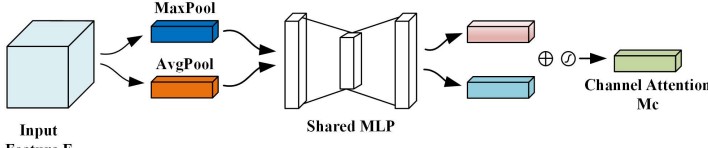

**Fig 4. Channel attention module.**

**Spatial Attention Module**

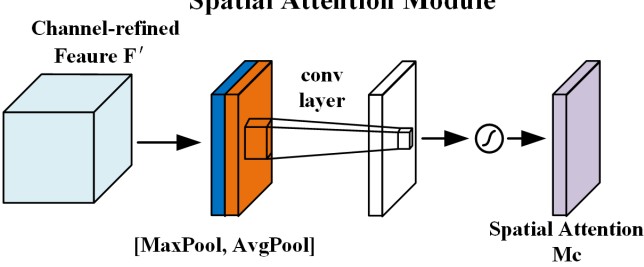

**Fig 5. Spatial attention module.**

to focus more effectively on informative features. The SENet module is lightweight, flexible, and contains relatively few parameters, making it suitable for integration into standard neural network architectures. The SE block adaptively learns relative feature weights based on the model's loss during training and applies corresponding reweighting, enabling the network to focus more effectively on informative features. By compressing along the spatial dimensions, the original multi-dimensional feature maps are squeezed into a one-dimensional vector. Global Average Pooling is then used to aggregate and encode global features, allowing the model to capture a degree of global receptive field. The interdependencies between channels are used to adaptively assess the importance of each feature channel, generating corresponding weights. The selected feature weights are first multiplied with their corresponding channels and then added back to the original feature map, achieving feature recalibration. The SE module performs a sequence of squeeze, excitation, and reweight operations, enabling the network to automatically learn feature importance. This enhances the model's ability to suppress redundant information, highlight critical features, and improve overall learning capacity.

In fire detection, both flame and smoke are non-rigid structures characterized by highly irregular and dynamic shapes. Key features of flames are primarily concentrated in their distinctive color distribution and edge contours, while smoke is defined by its semi-transparency and diffuse texture patterns. Moreover, forest fire monitoring is highly susceptible to interference and occlusion from complex and variable backgrounds. In the early stages of a fire, the target objects are typically very small in scale, which necessitates a strong multi-scale perception capability for effective detection.

In SSD, the detection of small objects relies on shallow feature maps, which, although high in resolution, have limited receptive fields and lack high-level semantic information. This often results in low classification confidence for small targets. Meanwhile, the deeper feature maps in SSD suffer from extremely low resolution due to repeated downsampling. Small objects typically occupy fewer than ten pixels in these maps, leading to significant loss of spatial information and, ultimately, inaccurate localization. These limitations negatively impact the precision of predicted bounding boxes, substantially lowering the Average Precision (AP). Additionally, the weak response of small objects in shallow feature maps often results in missed detections.

The channel attention mechanism enables the network to learn and dynamically adjust the weights of feature channels, significantly enhancing the representation of channels associated with flame color and smoke texture—both of which are critical for fire detection—while suppressing redundant channels related to background interference. The spatial attention mechanism generates a spatial weight map that allows for precise localization and emphasis of irregularly shaped and blurred fire regions within the image. It is particularly effective in improving the saliency of small or early-stage fire targets on the feature maps and suppressing interference from irrelevant background areas.

## 2.4 Improved algorithm flow chart

Manual patrols in forest fire prevention require substantial labor costs, and the difficulty increases significantly in areas with complex terrain. Currently, forest monitoring is typically conducted using a combination of device-based preliminary inspection and manual reinspection. However, relying on electronic devices alone for pre-inspection can lead to missed detections and false positives. To enhance the safety of the entire inspection area, it is essential to minimize the missed detection rate. Yet, a significant reduction in missed detections is often accompanied by an increase in false alarms. Due to the limited availability of data, this study adopts data augmentation techniques to provide the network with sufficient training samples, thereby optimizing network performance and improving accuracy. However, in the context of forest fires, even a small flame can cause irreparable damage. Therefore, in addition to improving precision, it is also critical to reduce the missed detection rate. To this end, a lightweight attention module is integrated into the already efficient SSD framework to enhance the network's focus on small objects and improve recall, without adding significant computational overhead. The algorithmic workflow is illustrated in Fig 6.

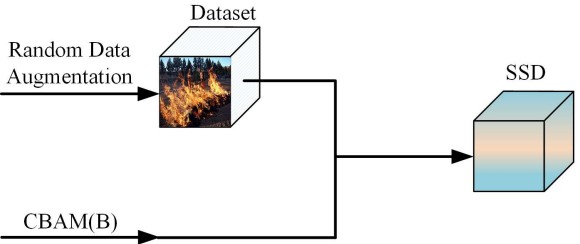

**Fig 6. Improved SSD flowchart.**

## 3. Experiment and result analysis

### 3.1 Dataset construction

In this study, a self-constructed fire detection dataset was used. The images were collected from real-world fire scenes around the globe using web scraping and video frame extraction techniques, ensuring both generalizability and scene diversity. The dataset does not include any samples from existing public datasets and has undergone strict quality filtering to retain only high-quality samples. All annotations were performed by a team of members trained through a standardized protocol, focusing on two object categories: "flame" and "smoke." A three-stage quality review process was conducted to ensure annotation accuracy and consistency. To align with the low-resolution characteristics of real-world applications and improve training efficiency, a resolution normalization process was applied: images with pixel dimensions under 1000 were retained at their original size, while those exceeding 1000 pixels were downscaled by a factor of 0.5. The dataset is organized in the PASCAL_VOC format and contains a total of 18,600 images. Following the standard machine learning paradigm, the data were divided into mutually exclusive subsets: 16,740 images were allocated for training and validation, used for model learning and hyperparameter tuning; 1,860 images were reserved as a strictly separated test set for evaluating generalization ability and final performance. The distribution of annotated objects in the training and test sets is illustrated in Fig 7.

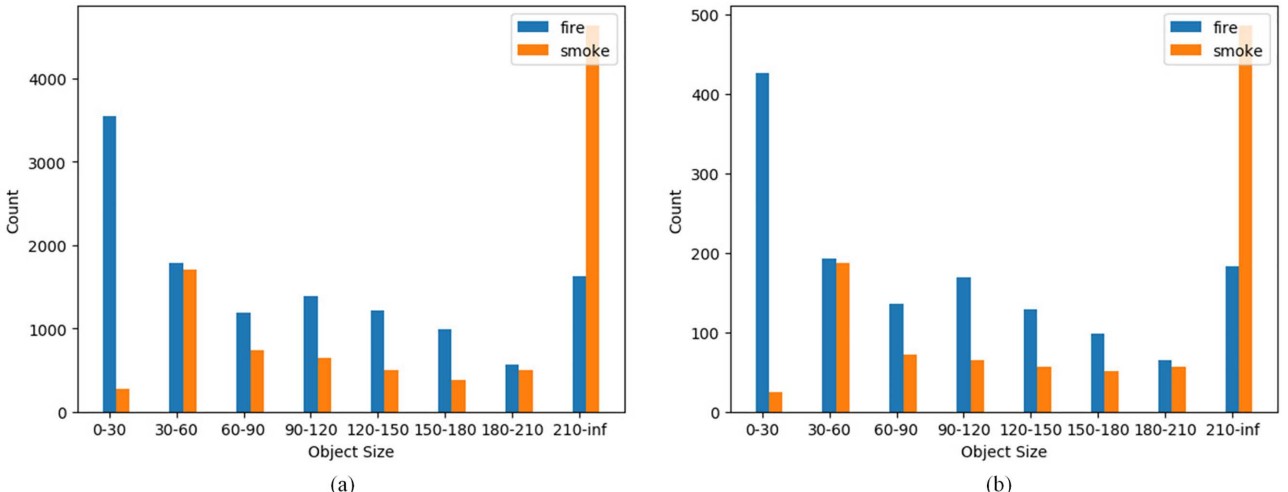

**Fig 7. Object count statistics.** Training set; (b) Test set. The horizontal axis represents the number of objects, and the vertical axis indicates the pixel area(unit: 1000 pixels) of the target objects.

## 3.2 Experimental environment and training parameters

The experiments were conducted on Ubuntu 18.04, using an Intel® Xeon® E5-2670 @ 2.60GHz CPU and an NVIDIA Tesla M40 GPU. The deep learning framework adopted was PyTorch 1.5.0 [24], which is widely used in the field of computer vision due to its flexibility, modularity, and extensive resource support. Python 3.6 was used as the programming language.

The input image size was set to 416×416 pixels. The learning rate was adjusted using the cosine annealing schedule [25], with a minimum learning rate of 0.00002 and a maximum of 0.002. The batch size was set to 32, and the random seed was fixed at 11 to ensure reproducibility. To achieve full model convergence, training was carried out for a total of 300 epochs, with each epoch taking approximately 13 minutes. The total training time was approximately 65 hours.

## 3.3 Model evaluation criteria

With the continued advancement of deep learning, numerous state-of-the-art algorithms have emerged. To identify the optimal model for fire detection, seven different models were evaluated. Considering the variability of flame and the interference introduced by smoke, both of which significantly influence model performance, this study employed a mixed training strategy using a unified training dataset that includes both flame and smoke images. Model performance was evaluated using the following metrics: Precision, Recall, Average Precision at Intersection over Union (IoU) threshold 0.5 (AP50), mean Average Precision at IoU 0.5 across multiple classes (mAP@0.5), and Frames Per Second (FPS). The threshold of 0.5 for IoU is chosen to ensure comparability between different models and research, as most models use 0.5 as the default IoU threshold.

The calculation methods for Precision and Recall are outlined in Equations 3 and 4, respectively. Specifically, True Positive (TP) refers to the number of samples where the IoU between the predicted bounding box and the ground truth bounding box is ≥ 0.5. False Positive (FP) represents the number of samples where the IoU between the predicted bounding box and the ground truth bounding box is < 0.5. False Negative (FN) denotes the number of samples where a real object is not detected by the model.

$$Precision = \frac{TP}{TP + FP} \tag{3}$$

$$Recall = \frac{TP}{TP + FN} \tag{4}$$

The calculation methods for AP and mAP are provided in Equations 5 and 6, respectively. For Recall ∈ {0, 0.1, 0.2,..., 1.0}, which includes 11 points, the maximum Precision at each higher Recall point is selected. The average of these 11 values is then computed to obtain AP. mAP is the mean of the AP values across all categories, where N represents the total number of categories, and AP_i denotes the average precision for the i-th category.

$$AP = \frac{1}{11} \sum_{r=0}^{10} \max_{\tilde{r} \geq r/10} P(\tilde{r}) \tag{5}$$

$$mAP = \frac{1}{N} \sum_{i=1}^{N} AP_i \tag{6}$$

The proposed improved algorithm was assessed using the same set of evaluation metrics to ensure fair comparison, and its performance was benchmarked against the originally best-performing model.

Since the test set and model weights remained unchanged throughout the evaluation, the reported results are consistent across multiple runs, yielding identical AP50 scores. As such, standard deviation or confidence intervals are not applicable in this evaluation.

### 3.4 Model convergence effect diagram

The final convergence behavior of the model is shown in Fig 8. To better utilize computational resources, improve training efficiency, and prevent early-stage weight disruption, the training process was divided into two phases. During the initial phase (first 150 epochs), frozen training was applied, where the backbone weights of the model were kept fixed, resulting in minimal parameter updates. After 150 epochs, the training transitioned to an unfrozen mode, allowing all network parameters to be updated. This transition from frozen to unfrozen training caused a noticeable change in parameter dynamics, leading to a temporary jump in the loss curve due to the shift in trainable parameters. As training progressed and network depth increased, the model continuously optimized, and the value of the cross-entropy loss gradually converged, indicating improved learning and representation capacity.

### 3.5 Experimental comparison and result analysis

According to the evaluation metrics described in Section 3.3, the SSD algorithm with VGG16 as the backbone consistently outperforms other conventional detection models across all four datasets, indicating its superior suitability for fire detection. As discussed in the Introduction, one-stage detectors generally offer faster detection speed compared to two-stage frameworks. On the other hand, lightweight front-end networks that trade accuracy for higher frame rates are also not ideal in this context. Considering these trade-offs, the original SSD model demonstrates a greater overall advantage. The relevant performance metrics are presented in Table 3. Therefore, this study focuses on optimizing the SSD architecture to develop a fire detection model that achieves both higher accuracy and better real-time performance.

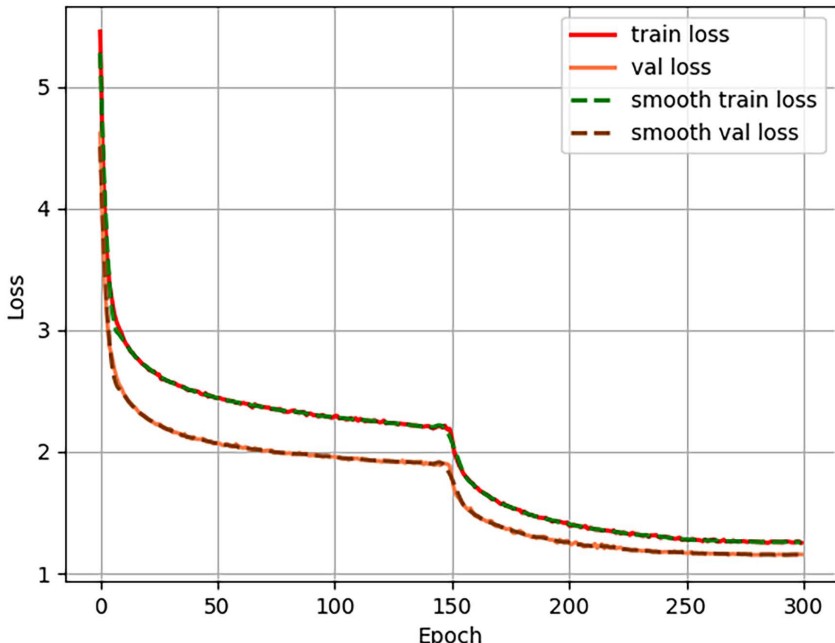

**Fig 8. Model convergence results.**

Firstly, based on the SSD algorithm, the dataset was modified by applying random data augmentation techniques to the training samples. As a result, the training set was expanded from 16,740 images to 25,000 images. This augmentation addressed the dataset's deficiency in small object instances and enriched the background information. As shown in Table 1, the detection accuracy improved after image augmentation. Compared to the original algorithm, this approach diversifies the data without altering the original samples, thereby increasing the utility of the images, enhancing detection accuracy, and improving the robustness and stability of model training.

Subsequently, the SSD algorithm was compared separately with the CBAM-SSD and SE-SSD, using the traditional VGG-16 structure as a backbone. An ablation study was conducted to assess the individual contributions of each attention mechanism and determine the optimal configuration that most effectively activates the network. To fully leverage pretrained weights, the backbone structure was left unchanged. Instead, attention modules were inserted after the auxiliary convolutional layers in the SSD framework. To analyze the impact of different modules and parameter scales on training performance, two insertion strategies were employed. For the CBAM module, one setup involved inserting the module after Conv7, while the other inserted it after both Conv7 and Conv10. The SE module was integrated using the same strategy, resulting in four comparative experiments. The experimental results are summarized in Table 2.

Through an exploration of the four attention-based training configurations, the results indicate that the SSD algorithm achieves optimal performance when two CBAM modules are integrated. Notably, the AP50 for flame detection increased by 1.42%. In contrast, SENet focuses solely on channel-wise attention, learning the relative importance of different channels while ignoring spatial variations within the same channel. As a result, when multiple instances of the same object class appear within a feature map, SENet struggles to distinguish their respective locations. CBAM, on the other hand, first applies channel attention to identify informative feature channels and then uses spatial attention to locate key regions within those channels. This hierarchical approach enables the network to more effectively learn and focus on critical information without introducing significant computational overhead. Consequently, the CBAM-based model is better suited for forest fire detection. The final network architecture incorporating the selected attention mechanism is illustrated in Fig 9.

Finally, the proposed approach integrates both data augmentation and attention mechanisms to form the final optimized model. By enriching the semantic content through data augmentation and subsequently incorporating attention modules, the network is better guided to focus on critical regions and capture the intrinsic relationships between data and features more effectively. The comparative performance results of this combined strategy are presented in Table 3.

**Table 1. Data augmentation before-and-after comparison.**

| Algorithm | mAP@0.5/% | AP50-fire/% | AP50-smoke/% |
|-----------|-----------|-------------|--------------|
| SSD | 96.02 | 93.60 | 98.44 |
| DA-SSD | 96.33 | 93.96 | 98.69 |

**Table 2. Contrast of different attention mechanisms.**

| Algorithm | mAP@0.5/% | AP50-fire/% | AP50-smoke/% | FPS |
|-----------|-----------|-------------|--------------|-----|
| SSD | 96.02 | 93.60 | 98.44 | 39.67 |
| SE-SSD(A) | 96.17 | 94.69 | 97.65 | 37.28 |
| SE-SSD(B) | 96.71 | 94.64 | 98.78 | 32.99 |
| CBAM-SSD(A) | 96.80 | 95.11 | 98.49 | 37.12 |
| CBAM-SSD(B) | 96.84 | 95.02 | 98.66 | 35.96 |

Note: A and B respectively represent the attention mechanism after adding the conv7 convolution layer and after adding the conv7 and conv10 convolution layers

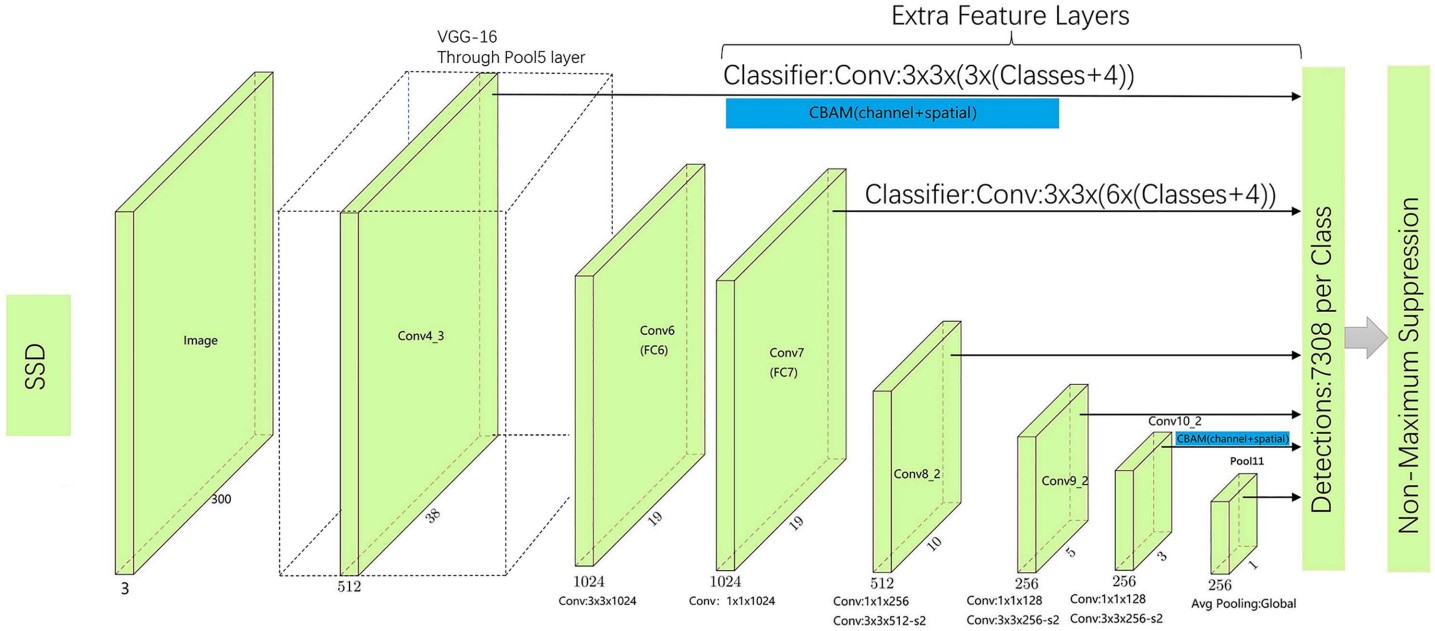

**Fig 9. CBAM-SSD model structure diagram.**

**Table 3. Comparison of target detection algorithms.**

| model | map@0.5/% | AP50-fire/% | AP50-smoke/% | FPS |
|---|---|---|---|---|
| YOLO-V5 [26] | 97.07 | 95.38 | 98.80 | 25.81 |
| Faster-R-CNN | 95.71 | 94.92 | 96.50 | 10.41 |
| Retina Net [27] | 96.91 | 94.84 | 98.97 | 29.00 |
| MobileNet-V2-SSD [28] | 94.54 | 92.12 | 96.96 | 54.21 |
| VGG-16-SSD | 96.02 | 93.60 | 98.44 | 39.67 |
| YOLO-V5-CBAM | 94.56 | 92.47 | 97.24 | 24.92 |
| Ours | 97.55 | 96.61 | 98.49 | 39.42 |

As shown in Table 3, the proposed CBAM-SSD effectively enhances the network's focus on critical information without introducing additional computational burden. This approach addresses the limitations of YOLOv5 and two-stage detectors, which struggle to meet real-time detection requirements, while also compensating for the accuracy deficiencies commonly observed in lightweight networks. Compared to the conventional SSD detection algorithm, the proposed method not only improves detection accuracy but also enhances the model's generalization ability through dataset augmentation. This enables robust performance across diverse fire detection scenarios and improves resistance to background interference. In contrast, the backbone and head of YOLOv5 are designed with a Cross-Stage Partial (CSP) structure that carefully balances speed and accuracy. Introducing CBAM into YOLOv5 disrupts its optimized information flow and feature extraction patterns, thereby degrading performance. Thus, the integration of CBAM into YOLOv5 proves counterproductive. The proposed CBAM-SSD also achieves a high frame processing rate, satisfying real-time requirements for forest fire monitoring. When embedded into various intelligent detection devices, it can serve as an effective solution for early warning and prevention of large-scale forest fires.

## 4. Conclusion and discussion

In this study, a forest fire detection algorithm named CBAM-SSD is proposed, which significantly improves detection accuracy without compromising inference speed. The method effectively addresses the challenges of detecting non-rigid objects such as flame and smoke while maintaining real-time performance. Random data augmentation was applied to enhance the training dataset, enabling the network to better learn the semantic features of flame and smoke despite a relatively small number of parameters. This enhances the model's generalization ability in forest fire detection. To further improve the detection accuracy of small objects, a dual-layer lightweight CBAM attention mechanism was integrated into the SSD architecture. This module suppresses redundant features and increases the sensitivity of the network to critical information. The attention mechanism helps overcome limitations of the original VGG-16-SSD, enhancing the network's perception of flame and smoke and enabling effective detection of early-stage fire events. Experimental results show that the proposed CBAM-SSD achieves a mAP@0.5 of 97.55%. Specifically, the AP50 for flame detection reaches 96.61%, an improvement of 3.01% over VGG-16-SSD, with a recall of 96.40%. For smoke detection, the AP50 reaches 98.49%, with a recall of 98.80%.

Although the proposed CBAM-SSD achieves real-time detection of forest fire regions with high precision and recall, it still has limitations in terms of model lightweighting, robustness, and hardware deployment. The limitation of CBAM-SSD essentially stems from the conflict between the "benefit of the attention mechanism" and the "cost of model complexity." In scenarios with sufficient computational power, abundant data, and complex objects, the precision gain from CBAM outweighs the efficiency loss, allowing for the insertion of multiple CBAM layers to improve accuracy. However, in resource-limited scenarios with limited data and simple objects, the efficiency loss and the risk of overfitting may surpass the precision gain. In such cases, targeted optimizations are necessary, such as inserting CBAM only into shallow feature maps critical for small object detection and using smaller convolution kernel sizes.

In practical applications, forest fire detection often requires deployment on UAVs due to the vast forest coverage. However, UAVs typically have limited computational resources, which necessitates the use of lightweight models. In future work, we plan to focus on reducing model complexity by redesigning an efficient backbone and a lightweight head to accelerate detection speed without sacrificing accuracy. We also aim to embed the optimized model into UAV platforms and develop a complete UAV-based inspection system to enhance its practical applicability. CBAM-SSD demonstrates the potential for high-precision, 24-hour continuous forest fire detection, offering valuable support for global forest fire prevention. Moreover, given the limited availability of forest fire datasets, our self-constructed dataset represents a valuable resource that we will leverage in future research to pursue further advancements in this field.

## Supporting information

**S1 File. Improved-ssd-pytorch.**
(ZIP)

## Acknowledgments

We deeply appreciate the constructive comments provided by the reviewers. Their valuable suggestions greatly contributed to the improvement of the manuscript and enhanced its overall quality.

## Author contributions

**Project administration:** Zhiwei Liu.

**Resources:** Shifeng Ruan.

**Validation:** Leilei Dong.

**Writing – original draft:** Diansheng Zhang.

**Writing – review & editing:** Yueyuan Zhang.

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
