## [Decision Letter · Decision Letter 0]

6 May 2025

Dear Dr. Zhang,

Thank you for submitting your manuscript to PLOS ONE. After careful consideration, we feel that it has merit but does not fully meet PLOS ONE’s publication criteria as it currently stands. Therefore, we invite you to submit a revised version of the manuscript that addresses the points raised during the review process.

We look forward to receiving your revised manuscript.

Kind regards,

Narendra Khatri, Ph.D.

Academic Editor

PLOS ONE

Journal Requirements:

3. We note that you have indicated that there are restrictions to data sharing for this study. PLOS only allows data to be available upon request if there are legal or ethical restrictions on sharing data publicly. For more information on unacceptable data access restrictions, please see http://journals.plos.org/plosone/s/data-availability#loc-unacceptable-data-access-restrictions .  

Additional Editor Comments :

Reviewers have evaluated your manuscript and recommended major revisions. Therefore, you are requested to revise the manuscript in accordance with the provided comments and resubmit it for further consideration

Reviewers' comments:

Reviewer's Responses to Questions

**Comments to the Author**

1. Is the manuscript technically sound, and do the data support the conclusions?

Reviewer #1: Partly

Reviewer #2: Partly

2. Has the statistical analysis been performed appropriately and rigorously?

Reviewer #1: Yes

Reviewer #2: No

3. Have the authors made all data underlying the findings in their manuscript fully available?

Reviewer #1: No

Reviewer #2: Yes

4. Is the manuscript presented in an intelligible fashion and written in standard English?

Reviewer #1: No

Reviewer #2: No

Reviewer #1: Research on Forest Fire Detection Based on Improved SSD Algorithm

General Comments

The manuscript, "Research on Forest Fire Detection Based on Improved SSD Algorithm," presents an innovative approach to fire detection in forest environments by improving the Single Shot MultiBox Detector (SSD) algorithm with a Convolutional Block Attention Module (CBAM). The study is well-structured, methodologically sound, and provides empirical evidence supporting the improved performance of the proposed model. The results demonstrate that the enhanced SSD model achieves superior accuracy, recall, and robustness compared to traditional fire detection algorithms. This research has practical significance for real-time fire monitoring in forested regions, which is a critical area in environmental conservation and disaster prevention. The study addresses an important real-world problem: the need for automated, real-time, and high-accuracy forest fire detection.

This is a well-conducted study with significant contributions to the field of deep learning-based fire detection. The proposed CBAM-SSD model improves fire detection accuracy and robustness, making it a valuable advancement. However, before publication, the major revisions are strongly recommended.

Specific Comments

• Title

o Consider specifying "deep learning-based detection" to make the title more informative. Suggested title revision: "Deep Learning-Based Forest Fire Detection Using an Improved SSD Algorithm with CBAM"

• Abstract

o The phrase "high real-time and embeddability required for detection are achieved" is unclear and grammatically awkward.

o The term "random data enhancement" is vague—it should specify data augmentation techniques used.

o The purpose of "compensate for constraints in the data collection process" is unclear—what constraints?

o "Constraint information" is unclear—does it mean feature extraction constraints?

o "Channel (Channel) and space (Spatial)" is redundant—just say "channel and spatial features."

o The phrase "serial structure is utilized" is vague—explain why this structure was chosen.

o The phrase "traditional detection algorithm" is too vague—specify which algorithm (e.g., VGG-16-SSD).

o The term "average accuracy AP" is redundant—just say "AP (Average Precision)".

o The recall rate of 96.40%—is it for fire detection, smoke detection, or both?

o mAP (97.55%)—better to clarify that it is the mean average precision across all categories.

o "Lower object detection false alarm" → should be "lower false alarm rate."

o "Miss rate" should specify the decrease in false negatives.

o "Satisfies the requirements" → what requirements? Who sets them?

• Introduction

o Lack of Clear Justification for SSD Selection – The introduction does not explain why SSD was chosen over YOLO or Faster R-CNN.

o Missing Citations – Several references are missing or improperly formatted (e.g., "Error! Reference source not found.").

o Give the abbreviations used at the beginning of the sentence.(ex, SSD, LAM, etc)

o No explanation of YOLOv4-Tiny's weaknesses – Why was SSD chosen instead? Missing comparison with SSD – The introduction should mention how SSD differs from YOLO.

o Unclear Explanation of Prior Work – The discussion of existing methods lacks coherence and should include a comparison table summarizing strengths and weaknesses.

o Overly Technical Terms Introduced Too Early – Terms like CBAM, anchor boxes, feature pyramid appear without sufficient explanation.

• Experiment and result analysis

o The "Experiment and Result Analysis" section is crucial for demonstrating the effectiveness of the proposed CBAM-SSD model. This section presents strong experimental validation but needs more details on dataset biases, training choices, computational cost, and real-world deployment feasibility.

o Missing details on dataset composition – Where was the fire data collected? Where were these fire images taken? From real-world fires or simulations? Were there biases in the dataset?- No discussion on dataset bias – Are there regional, seasonal, or environmental biases?

o Lack of justification for hyperparameters – Why was 300 epochs chosen? How was the learning rate tuned?

o Image resizing details unclear – Why was 1000 pixels chosen as the threshold?

o No mention of training time per epoch – How long did the full training process take?

o No explanation of AP calculation – Did it use IoU thresholds (e.g., 0.5, 0.75, or averaged over multiple IoUs)?

o Insufficient explanation of ablation studies – Needs more discussion on why CBAM outperforms SE-Net.

o Computational efficiency not discussed – Does CBAM-SSD increase inference time compared to traditional SSD?

o Comparison with real-world deployment – How would this model perform on edge devices (e.g., drones, mobile applications)?

• Conclusion and discussion

o The "Conclusion and Discussion" section should provide a strong summary of findings, limitations, real-world implications, and future research directions. While the manuscript presents a well-structured conclusion, several areas require improvements.

o The conclusion should be more impactful – It lacks a strong final statement on the model’s real-world benefits.

o "Dual perception mechanism" is unclear – Does this refer to channel and spatial attention in CBAM?

o No discussion on real-world deployment challenges – How does this model perform in edge devices, UAVs, or surveillance cameras?

o Limitations are not fully addressed – What are the trade-offs of using CBAM-SSD?

o Future work is too vague – The section should mention specific techniques for optimizing the model (e.g., quantization, federated learning).

o Lacks a strong concluding remark – The last sentence should emphasize impact and real-world significance.

• Reference

o Redundant or outdated references – Some citations refer to older versions of algorithms (e.g., YOLOv4 instead of YOLOv8).

Reviewer #2: 1. Writing Quality and Structure - The manuscript contains numerous grammatical, typographical, and formatting errors, such as:

“Error! Reference source not found.” — appearing multiple times suggests broken citations from a word processor.

Long and unclear sentences, e.g., "The mapping space and channel feature information can be aggregated..." lacks precision and clarity.

Recommend to replace broken references and clean up figure labels and captions. Each figure should be referenced in the body text and explained.

2. The paper reports only single-point values (e.g., mAP = 97.55%) without any standard deviation, variance, or confidence intervals. This makes it difficult to assess the statistical significance of the reported improvements, especially when the gain is marginal (e.g., a 1.53% improvement over baseline).

There is no use of statistical tests (e.g., paired t-tests, Wilcoxon signed-rank test) to evaluate whether differences between models are statistically meaningful.

Suggestion - Include standard deviation or confidence intervals for key metrics. Clearly state how many trials were run and whether the results are averaged. Apply statistical tests to support claims of superiority when differences are marginal.

3. The dataset construction lacks detail—while the authors mention it was compiled using web crawlers and simulation, there is no discussion of ground truth annotation quality, data diversity, or whether any public datasets were included.

4. The core techniques—CBAM, data augmentation, and SSD improvements—have been widely used and studied in various domains. The novelty lies primarily in their application to forest fire detection, but the paper does not explicitly justify why this combination is uniquely effective for that use case.

Suggestions: Provide deeper insight into why CBAM specifically improves fire/smoke detection (e.g., are flame patterns especially sensitive to spatial context?).

Consider benchmarking against CBAM + YOLOv5 or CBAM + RetinaNet to isolate the benefit of your integration with SSD versus other modern detectors.

Additionally, include a section highlighting novel insights or unique domain challenges to strengthen your contribution claim.

**Do you want your identity to be public for this peer review?** For information about this choice, including consent withdrawal, please see our Privacy Policy

Reviewer #1: **Yes: ** Adinugroho

Reviewer #2: No

---

## [Author Response · Author response to Decision Letter 1]

17 Jun 2025

Dear Editors and Reviewers:

Thank you for your letter and for the reviewers’ comments concerning our manuscript entitled “Research on forest fire detection based on improved SSD algorithm” (ID: PONE-D-24-53870). We greatly appreciate the valuable and constructive comments provided by the reviewers and editor, which have been instrumental in improving the quality and clarity of our manuscript. We have carefully addressed each point and revised the manuscript accordingly.

Response to Narendra Khatri, Ph.D:

1. We have revised the manuscript in accordance with the formatting requirements of PLOS ONE. We carefully reviewed all the comments provided by the reviewers and have made the corresponding revisions in the manuscript. In addition, we have corrected grammatical issues throughout the entire text.

2. In response to PLOS ONE request, we have also uploaded the source code to GitHub�https://github.com/zhangdiansheng1999/fire-detection.git

3. Our dataset was self-constructed and is owned by the Key Laboratory of RF Communication and Sensor Network in Jiangxi Province, China. The laboratory does not allow open distribution. Interested researchers may request access by contacting Professor Liu at: ecjtu-tech@outlook.com.

Response to Reviewers:

Reviewer #1:

• Title

o Consider specifying "deep learning-based detection" to make the title more informative. Suggested title revision: "Deep Learning-Based Forest Fire Detection Using an Improved SSD Algorithm with CBAM"

The author’s answer:

1) We have modified the title to: "Deep Learning-Based Forest Fire Detection Using an Improved SSD Algorithm with CBAM".

• Abstract

o The phrase "high real-time and embeddability required for detection are achieved" is unclear and grammatically awkward.

o The term "random data enhancement" is vague—it should specify data augmentation techniques used.

o The purpose of "compensate for constraints in the data collection process" is unclear—what constraints?

o "Constraint information" is unclear—does it mean feature extraction constraints?

o "Channel (Channel) and space (Spatial)" is redundant—just say "channel and spatial features."

o The phrase "serial structure is utilized" is vague—explain why this structure was chosen.

o The phrase "traditional detection algorithm" is too vague—specify which algorithm (e.g., VGG-16-SSD).

o The term "average accuracy AP" is redundant—just say "AP (Average Precision)".

o The recall rate of 96.40%—is it for fire detection, smoke detection, or both?

o mAP (97.55%)—better to clarify that it is the mean average precision across all categories.

o "Lower object detection false alarm" → should be "lower false alarm rate."

o "Miss rate" should specify the decrease in false negatives.

o "Satisfies the requirements" → what requirements? Who sets them?

The author’s answer:

1) Based on your suggestions and a summary of the manuscript, we have reorganized the abstract to highlight the challenges in forest fire detection, the adopted technical approach, experimental results, and contributions. The modifications in the abstract are marked in red in the revised version.

2) We removed the phrase "high real-time and embeddability required for detection are achieved" and further summarized the problem of forest fire detection: the varying scales and complex features of flame and smoke, as well as false positives and missed detections caused by environmental interference.

3) We have changed "random data enhancement" to data augmentation techniques and added the specific methods used.

4) We have revised "compensate for constraints in the data collection process" to: address problems such as insufficient and incomplete data during the collection process.

5) Regarding "Channel (Channel) and space (Spatial)", since we added the channel attention mechanism first and then the spatial attention mechanism to the SSD, we referred to it as a serial structure. This part has now been rewritten.

6) According to your suggestion, we changed "traditional detection algorithm" to baseline SSD.

7) Yes, we have changed "average accuracy AP" to AP50, and "mAP" to mAP@0.5. We have also clarified that the mAP@0.5 refers to the mAP@0.5 across all categories.

8) Since no unified standard has been set for forest fire detection, we revised the sentence to: provide an efficient, convenient, and accurate solution for forest fire detection.

• Introduction

o Lack of Clear Justification for SSD Selection – The introduction does not explain why SSD was chosen over YOLO or Faster R-CNN.

o Missing Citations – Several references are missing or improperly formatted (e.g., "Error! Reference source not found.").

o Give the abbreviations used at the beginning of the sentence.(ex, SSD, LAM, etc)

o No explanation of YOLOv4-Tiny's weaknesses – Why was SSD chosen instead? Missing comparison with SSD – The introduction should mention how SSD differs from YOLO.

o Unclear Explanation of Prior Work – The discussion of existing methods lacks coherence and should include a comparison table summarizing strengths and weaknesses.

o Overly Technical Terms Introduced Too Early – Terms like CBAM, anchor boxes, feature pyramid appear without sufficient explanation.

The author’s answer:

1) We used Word’s cross-reference function for citing references. When we modified the references, we failed to update the cross-referenced numbers in the main text, which led to such errors. We have corrected all reference formats in the manuscript. We apologize for our oversight.

2) According to your suggestion, we have added content in the introduction to explain YOLO, Faster R-CNN, and SSD, their respective advantages and disadvantages, and why we chose SSD. We have also added a paragraph introducing the structure of the paper.

• Experiment and result analysis

o The "Experiment and Result Analysis" section is crucial for demonstrating the effectiveness of the proposed CBAM-SSD model. This section presents strong experimental validation but needs more details on dataset biases, training choices, computational cost, and real-world deployment feasibility.

o Missing details on dataset composition – Where was the fire data collected? Where were these fire images taken? From real-world fires or simulations? Were there biases in the dataset?- No discussion on dataset bias – Are there regional, seasonal, or environmental biases?

o Lack of justification for hyperparameters – Why was 300 epochs chosen? How was the learning rate tuned?

o Image resizing details unclear – Why was 1000 pixels chosen as the threshold?

o No mention of training time per epoch – How long did the full training process take?

o No explanation of AP calculation – Did it use IoU thresholds (e.g., 0.5, 0.75, or averaged over multiple IoUs)?

o Insufficient explanation of ablation studies – Needs more discussion on why CBAM outperforms SE-Net.

o Computational efficiency not discussed – Does CBAM-SSD increase inference time compared to traditional SSD?

o Comparison with real-world deployment – How would this model perform on edge devices (e.g., drones, mobile applications)?

The author’s answer:

1) We have added Section 3.1 to explain the dataset in detail based on your comments. A figure showing the distribution of target numbers in the dataset has also been added.

2) We chose 300 epochs to ensure that the model converges sufficiently and achieves optimal performance. The learning rate was adjusted using cosine annealing, with a minimum value of 0.00002 and a maximum value of 0.002. We have described this in Section 3.2 of the revised manuscript.

3) Due to our oversight, we failed to mention the IoU threshold. In our experiments, the IoU threshold was always set to 0.5. This has now been clarified in Section 3.3.

4) We added a discussion in Section 3.4 explaining why CBAM outperforms SE-Net.

5) Compared with VGG-16-SSD, the inference time of our model remains nearly unchanged. As shown in Table 3, the FPS of VGG-16-SSD is 39.76, and that of our model is 39.42.

6) Since we have not yet deployed the model on UAVs, we have not conducted a comparison. Our next plan is to deploy the model on UAVs.

• Conclusion and discussion

o The "Conclusion and Discussion" section should provide a strong summary of findings, limitations, real-world implications, and future research directions. While the manuscript presents a well-structured conclusion, several areas require improvements.

o The conclusion should be more impactful – It lacks a strong final statement on the model’s real-world benefits.

o "Dual perception mechanism" is unclear – Does this refer to channel and spatial attention in CBAM?

o No discussion on real-world deployment challenges – How does this model perform in edge devices, UAVs, or surveillance cameras?

o Limitations are not fully addressed – What are the trade-offs of using CBAM-SSD?

o Future work is too vague – The section should mention specific techniques for optimizing the model (e.g., quantization, federated learning).

o Lacks a strong concluding remark – The last sentence should emphasize impact and real-world significance.

The author’s answer:

1) According to your suggestion, we have revised the conclusion.

2) The term "dual perception mechanism" refers to the addition of two CBAM modules in the SSD. As this expression may cause misunderstanding, we have changed "dual perception mechanism" to "mechanism".

3) We have not yet deployed the model on edge devices, UAVs, or surveillance cameras. Such deployment requires integration with device systems and construction of deployment platforms. We plan to implement this in the next stage of our work.

4) The unique advantages of CBAM-SSD in forest fire detection are discussed in Section 1.3. In addition, we summarized the disadvantages of CBAM-SSD in the conclusion: one limitation is that deployment on edge devices, UAVs, or surveillance cameras requires model lightweighting, as these devices lack sufficient computing power.

• Reference

o Redundant or outdated references – Some citations refer to older versions of algorithms (e.g., YOLOv4 instead of YOLOv8).

The author’s answer:

1) We have corrected the reference formats throughout the manuscript. We apologize for our oversight.

Reviewer #2:

1. Writing Quality and Structure - The manuscript contains numerous grammatical, typographical, and formatting errors, such as:

“Error! Reference source not found.” — appearing multiple times suggests broken citations from a word processor.

Long and unclear sentences, e.g., "The mapping space and channel feature information can be aggregated..." lacks precision and clarity.

Recommend to replace broken references and clean up figure labels and captions. Each figure should be referenced in the body text and explained.

The author’s answer:

1) We have restructured the entire manuscript and improved grammatical correctness throughout.

2) We used the cross-reference function in Word for citations. When modifying the reference list, we failed to update the in-text reference numbers, resulting in errors like "Error! Reference source not found." We have now corrected all reference formats. We sincerely apologize for our carelessness.

3) According to the PLOS ONE submission guidelines, figure labels and captions should be retained in the manuscript. We followed these requirements. However, if this is considered inappropriate, we are willing to remove them.

2. The paper reports only single-point values (e.g., mAP = 97.55%) without any standard deviation, variance, or confidence intervals. This makes it difficult to assess the statistical significance of the reported improvements, especially when the gain is marginal (e.g., a 1.53% improvement over baseline).

There is no use of statistical tests (e.g., paired t-tests, Wilcoxon signed-rank test) to evaluate whether differences between models are statistically meaningful.

Suggestion - Include standard deviation or confidence intervals for key metrics. Clearly state how many trials were run and whether the results are averaged. Apply statistical tests to support claims of superiority when differences are marginal.

The author’s answer:

1) Thank you for your valuable suggestion. We also agree that including standard deviation or statistical testing is typically important for evaluating the significance of performance differences.

2) However, in our case, the reported AP50 values were obtained using a deterministic model on a fixed test set. We set fixed random seeds during training to avoid randomness, and disabled data augmentation during testing to ensure realism. Therefore, repeated evaluations produce identical results, and statistical testing is not applicable in this situation. We have added clarification regarding this in Section 3.3 of the revised manuscript.

3. The dataset construction lacks detail—while the authors mention it was compiled using web crawlers and simulation, there is no discussion of ground truth annotation quality, data diversity, or whether any public datasets were included.

The author’s answer:

1) We have added a dedicated section (Section 3.1) to explain our dataset source and data quality.

2) In addition, we included a figure showing the target distribution of the dataset to help illustrate its composition.

4. The core techniques—CBAM, data augmentation, and SSD improvements—have been widely used and studied in various domains. The novelty lies primarily in their application to forest fire detection, but the paper does not explicitly justify why this combination is uniquely effective for that use case.

Suggestions: Provide deeper insight into why CBAM specifically improves fire/smoke detection (e.g., are flame patterns especially sensitive to spatial context?).

The author’s answer:

1) Thank you for your suggestion. In Section 1.3, we added three new paragraphs discussing the characteristics of fire detection, the limitations of SSD, and the advantages of CBAM.

5. Consider benchmarking against CBAM + YOLOv5 or CBAM + RetinaNet to isolate the benefit of your integration with SSD versus other modern detectors.

The author’s answer:

1) We added CBAM to YOLOv5, and the results are presented in Table 3. We also provided analysis of YOLOv5 with CBAM.

6. Additionally, include a section highlighting novel insights or unique domain challenges to strengthen your contribution claim.

The author’s answer:

1) According to your suggestion, we have revised the conclusion and added a discussion of novel insights and domain-specific challenges.

We deeply appreciate the reviewers’ and editor’s insightful comments and suggestions, which have been instrumental in refining our work. We hope the revised manuscript will be considered suitable for publication in PLOS ONE.

Sincerely,

Yueyuan Zhang

School of Information and Software Engineering,

East China Jiaotong University, Nanchang, China.

Email: zyyaney1981@hotmail.com

---

## [Decision Letter · Decision Letter 1]

5 Aug 2025

Dear Dr. Zhang,

Thank you for submitting your manuscript to PLOS ONE. After careful consideration, we feel that it has merit but does not fully meet PLOS ONE’s publication criteria as it currently stands. Therefore, we invite you to submit a revised version of the manuscript that addresses the points raised during the review process.

We look forward to receiving your revised manuscript.

Kind regards,

Narendra Khatri, Ph.D.

Academic Editor

PLOS ONE

Journal Requirements:

Additional Editor Comments :

Minor Revision

Reviewers' comments:

Reviewer's Responses to Questions

**Comments to the Author**

Reviewer #1: All comments have been addressed

2. Is the manuscript technically sound, and do the data support the conclusions?

Reviewer #1: Yes

3. Has the statistical analysis been performed appropriately and rigorously?

Reviewer #1: Yes

4. Have the authors made all data underlying the findings in their manuscript fully available?

Reviewer #1: Yes

5. Is the manuscript presented in an intelligible fashion and written in standard English?

Reviewer #1: Yes

Reviewer #1: The authors have made significant revisions based on the first round of review, which have substantially strengthened the manuscript.

• The authors have addressed the suggestion, and the revised title is clear and descriptive.

• The authors have made appropriate changes based on the suggestions. The abstract is now clearer, with terms more specifically defined and phrasing made more precise.

• The authors have satisfactorily addressed some points on first round, improving the clarity and structure of the introduction.

• The authors have made good progress in addressing first round concerns on Experiment and Result Analysis. They provided additional details on dataset construction and training, though further analysis of deployment feasibility (e.g., edge devices) might be useful in future versions.

• The conclusion is now more impactful, and the authors have clearly addressed the feedback regarding the "dual perception mechanism" and real-world deployment challenges. The mention of UAV deployment plans is a good addition.

• The authors have appropriately addressed the issues with citations.

Overall, the revisions have significantly strengthened the manuscript, and it appears ready for publication with minor adjustments related to :

• Several abbreviations are introduced without being defined or explained when first mentioned in the manuscript. Generally, it's best to avoid using abbreviations in the abstract of a paper, especially if the audience may not be familiar with them. However, if you must use abbreviations, spell out the full term followed by the abbreviation in parentheses the first time it's used, and then you can use the abbreviation throughout the rest of the abstract and the paper.

• The authors should explicitly state the calculation methods for performance metrics, especially mAP, and provide more context on how they chose thresholds like IoU (0.5) or AP50 (from reference?). A brief explanation of the metric calculation process would help the reader understand the significance of the results.

• Although the authors mention some limitations in the conclusion, the discussion could benefit from more depth, particularly in terms of trade-offs associated with using CBAM-SSD (Trade-offs between accuracy and computational efficiency? Potential overfitting?)

• Real-world application and deployment. It would be beneficial to see further exploration of real-world deployment, particularly regarding the integration of the model into UAVs and edge devices, as mentioned in the conclusion.

**Do you want your identity to be public for this peer review?** For information about this choice, including consent withdrawal, please see our Privacy Policy

Reviewer #1: No

---

## [Author Response · Author response to Decision Letter 2]

7 Aug 2025

Dear Editors and Reviewers:

Thank you for your letter and for the reviewers’ comments concerning our manuscript entitled “Deep learning-based forest fire detection using an improved SSD algorithm with CBAM” (ID: PONE-D-24-53870). We greatly appreciate the valuable and constructive comments provided by the reviewers and editor, which have been instrumental in improving the quality and clarity of our manuscript. We have carefully addressed each point and revised the manuscript accordingly.

Response to Narendra Khatri, Ph.D:

1. The reference list cited in this paper does not include any retracted papers.

2. We have identified a spelling error in the evaluation criteria in Table 2. The correction has been made and highlighted in red in the "Revised Manuscript with Track Changes" file.

3. We have re-stated the financial disclosure in the Cover Letter.

4. We sincerely appreciate your guidance and your patience in providing assistance.

Response to Reviewers:

Reviewer #1:

Question1

• Several abbreviations are introduced without being defined or explained when first mentioned in the manuscript. Generally, it's best to avoid using abbreviations in the abstract of a paper, especially if the audience may not be familiar with them. However, if you must use abbreviations, spell out the full term followed by the abbreviation in parentheses the first time it's used, and then you can use the abbreviation throughout the rest of the abstract and the paper.

The author’s answer:

We sincerely apologize for not identifying this issue during multiple rounds of review. We have now spelled out the full terms for abbreviations such as Faster R-CNN and SPPNet the first time they appear.

Question2

• The authors should explicitly state the calculation methods for performance metrics, especially mAP, and provide more context on how they chose thresholds like IoU (0.5) or AP50 (from reference?). A brief explanation of the metric calculation process would help the reader understand the significance of the results.

The author’s answer:

In Section 3.3, we provide a detailed explanation of the calculation methods for Precision, Recall, AP, and mAP, and we also explain why we use 0.5 as the threshold for IoU.

Question3

• Although the authors mention some limitations in the conclusion, the discussion could benefit from more depth, particularly in terms of trade-offs associated with using CBAM-SSD (Trade-offs between accuracy and computational efficiency? Potential overfitting?)

• Real-world application and deployment. It would be beneficial to see further exploration of real-world deployment, particularly regarding the integration of the model into UAVs and edge devices, as mentioned in the conclusion.

The author’s answer:

We have divided the conclusion into three parts: the achievements of CBAM-SSD, the limitations of CBAM-SSD, and the future directions of CBAM-SSD’s application. In the second part, we focus on discussing how to balance precision and computational efficiency when using CBAM-SSD.

We deeply appreciate the reviewers’ and editor’s insightful comments and suggestions, which have been instrumental in refining our work. We hope the revised manuscript will be considered suitable for publication in PLOS ONE.

Sincerely,

Yueyuan Zhang

School of Information and Software Engineering,

East China Jiaotong University, Nanchang, China.

Email: zyyaney1981@hotmail.com

---

## [Editor Report · Decision Letter 2]

17 Sep 2025

Deep learning-based forest fire detection using an improved SSD algorithm with CBAM

PONE-D-24-53870R2

Dear Dr. Zhang,

We’re pleased to inform you that your manuscript has been judged scientifically suitable for publication and will be formally accepted for publication once it meets all outstanding technical requirements.

Kind regards,

Narendra Khatri, Ph.D.

Academic Editor

PLOS ONE

Additional Editor Comments (optional):

Accepted
---

## [Editor Report · Acceptance letter]

PONE-D-24-53870R2

PLOS ONE

Dear Dr. Zhang,

I'm pleased to inform you that your manuscript has been deemed suitable for publication in PLOS ONE. Congratulations! Your manuscript is now being handed over to our production team.

Kind regards,

on behalf of

Dr. Narendra Khatri

Academic Editor

PLOS ONE